# Impacts of COVID-19 on Agricultural Production Branches: An Investigation of Anxiety Disorders among Farmers

**Celal Cevher** [1,*] ID **, Bulent Altunkaynak** [2] ID **and Meltem Gürü** [3]

1 Field Crops Central Research Institute, Şehit Cem Ersever 9-11, Yenimahalle, 06170 Ankara, Turkey
2 Department of Statistics, Faculty of Science, Gazi University, Beşevler, 06500 Ankara, Turkey; bulenta@gazi.edu.tr
3 Health Care Center, Gazi University Rectorate Campus, Beşevler, 06500 Ankara, Turkey; meltemguru@gmail.com
* Correspondence: celal.cevher@tarimorman.gov.tr

**Abstract:** The aim of our study was to determine the level of anxiety among farmers in different agricultural branches in Turkey during the COVID-19 outbreak and to examine its association with socioeconomic concerns and social support variables. Based on a survey of 2125 Turkish farm enterprises, this study examined effects on agricultural production during the implementation of COVID-19 restrictions among agricultural branches. The Generalized Anxiety Disorder Scale and Oslo Social Support Scale were used in the study. Survey data were collected from farmers by phone. Age, place of residence, income status, agricultural branches, land size, the use of trucks, animal husbandry, access to technical support from agricultural organizations, access to support from neighbors, and social support level were found to have a significant effect on anxiety level ($p < 0.05$). The anxiety levels of farmers engaged in animal husbandry and vegetable farming were found to be higher than those in other agricultural branches. The lowest anxiety level was observed in farmers engaged in cereal production. Taking these results into consideration is important for preventing problems in agricultural production. If anxiety levels are not improved, it is predicted that farmers in agricultural production branches with high levels of anxiety will move towards branches with lower levels of anxiety.

**Keywords:** agricultural branches; COVID-19; farmers' anxiety; turkey

## 1. Introduction

After it was first reported in Wuhan in December 2019, the novel Coronavirus Disease (COVID-19) spread rapidly throughout the world, causing the deaths of many people. The World Health Organization (WHO) declared this epidemic to be a pandemic on 11 March 2020 [1]. Since the first report on 11 March 2020, various health and safety precautions have been taken in Turkey in order to prevent the spread of the disease. First, education was suspended in all schools and universities, many collective organizations such as conferences and congresses were postponed, intercity travel was subject to the permission of the governor, a curfew was declared for individuals over 65 and those under 18, and then curfews at certain intervals were established for all individuals living in 30 metropolitan cities [2].

These measures, which were strictly carried out between March and May 2020, had been extended as of June 2020. Although these measures were taken to protect the public from the epidemic, the mental health of many people has been adversely affected as a result of both epidemic anxiety and the restrictive lifestyle imposed by measures to control the spread of the disease [3]. It has been reported that psychiatric illnesses, especially anxiety disorders, have increased during the epidemic [4]. COVID-19 and the restrictive measures towards containing the spread of infections have seriously affected agricultural production branches and jeopardized food security [5]. The rapid spread of the coronavirus disease

(COVID-19) in Turkey has prompted the quick implementation of disease containment and other COVID-19 response measures. These courses of action have resulted in the reduction in the agricultural workforce, declining household incomes, rising rates of unemployment, and disruptions in agricultural supply chains, which have had severe impacts on Turkey's food security situation.

With the spread of COVID-19, the world economy as a whole was negatively affected, and production was disrupted in many business sectors [4]. The agricultural sector was also affected by these disruptions. In some regions, reductions in agricultural production are expected due to the passing of planting time and the imposition of restrictions, and employees in the sector face the inevitable risk of low income [6]. Due to its nature, agricultural production is always exposed to higher risks than other branches (climatic conditions, natural disasters, labor, etc.). Agricultural production is known to be a sector with a high-risk rate compared to other sectors, even in pre-pandemic times [7]. Therefore, farmers are experiencing heavy economic consequences of the pandemic period. COVID-19 and the restrictive measures towards containing the spread of its infections have seriously affected the agricultural workforce and jeopardized food security [8].

Studies have shown that people with serious economic losses during the pandemic have become more vulnerable to mental health problems [9]. Although there are studies examining the economic effects of the COVID-19 outbreak on farmers, studies on the psychological effects are limited. In a study examining a single case [10], the authors investigated the suicide of a farmer in India due to socioeconomic problems that occurred during the pandemic. Moreover, prior to the COVID-19 outbreak, a study [11] found that farmers had higher levels of anxiety than the general population and were at risk of developing psychiatric diseases.

Studies investigating the psychological effects of the COVID-19 outbreak seem to be focused on the general population, patients, healthcare professionals, children, seniors, or university students [12–14]. Studies on farmers are limited. Farmers who already live in poor socioeconomic conditions are thought to be vulnerable to the psychological effects of COVID-19 [10]. It is important to examine the psychological conditions of those at the forefront of food and agriculture and to take evidence-based measures to keep the general public safe. Therefore, to maintain food security during the pandemic, farmers' health should be treated as a serious issue in order to ensure that it does not affect their productivity [15]. The social and economic impacts of COVID-19 have been reported by researchers that it can have wide-ranging negative effects on human well-being [16,17]. Darnhofer reported that the factors affecting farmers resistance in their study during the COVID-19 pandemic [18]. Darnhofer suggested that the process-relational approach displaces the presumption of structural determination and thus allows to highlight the ever-present openings for change [19].

Sustainable agriculture is becoming increasingly important in the world. In this context, studies dealing with the environmental, economic, and social effects of sustainable agriculture are gradually increasing [20]. In terms of ensuring economic sustainability in rural areas, studies related to the development of rural tourism [21] and animal husbandry [22] have been performed. Sustainable agriculture is possible with the right agricultural incentives and innovative research [23]. It is likely that not only the healthcare field but also the sustainable agriculture area will be affected by COVID-19.

The hypothesis of the study is that farmers with strong social support will experience less anxiety than farmers with poor social support. The results of this study are also important in terms of identifying the farmers, who form the basis of the food supply chain, without experiencing more severe mental disorders. In this way, farmers whose mental health is affected can be identified in advance and social and economic support can be given earlier. Providing economic and social support early is an important step in helping farmers return to normal life.

In this context, the aim of this study was to determine the level of anxiety of farmers in Turkey during the COVID-19 outbreak and to examine the effects of social support and socioeconomic variables.

## 2. Materials and Methods

### 2.1. Study Population and Sample

COVID-19 restrictions were applied simultaneously in all provinces in Turkey. The population of this study consisted of farmers in Turkey. According to the 2019 Farmer Registration System, the number of farmers in Turkey is 2,264,000. A total of 2125 farmers from 22 cities were randomly selected using multi-stage cluster sampling. Turkey has different ecological and production patterns in terms of geographic region. The names of the farmers to be interviewed were obtained by contacting the official and agricultural organizations in the selected provinces. Before starting the survey interview, farmers were informed about the importance of working. After the information given, they were asked whether they would participate in the survey or not. The survey has been completed with farmers who answered ''yes''. The selected provinces are shown on the map in Figure 1. Data were collected in May using a telephone survey, as the highest occurrence of death was recorded in May. May was also a month in which preventive measures were implemented. Furthermore, 41,550 people were infected with the disease and 1282 people died that month [24].

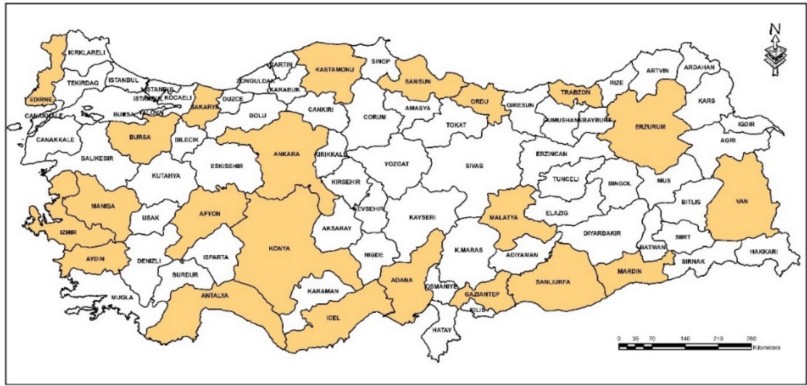

**Figure 1.** Map of the study area.

### 2.2. Data and Survey

A sociodemographic data form, the Generalized Anxiety Disorder Scale, and the Oslo Social Support Scale were used in this study.

A broad-based questionnaire was designed to investigate the risk perception and determinants in agricultural branches. The final version of the questionnaire was reviewed by agribusiness, agricultural economics, rural sociologists, medical psychologists, and researchers. Next, a preliminary questionnaire test was conducted on five farmers in agriculture to determine the appropriateness of the order and flow of questions and the clarity of the statements. At the beginning of each interview, the respondents were given a brief introduction to the study and were asked to answer questions regarding the potential consequences of the pandemic on the business activities of their agricultural branches.

The questionnaire contained structured items consisting of (1) characteristics of the surveyed agricultural branches; (2) production changes caused by COVID-19 restrictions in agricultural branches; and (3) the level of anxiety disorder of farmers. The survey interviews were held between 15 April and 30 May.

Related agriculture branches in the study are characterized as follows.

Cereal farming: human nutrition, animal feed requirements, and farmers' income source.

Cereal + animal farming: small business, low income, and very common in Turkey.

Fruit farming: favorable climatic conditions, important export product, and good source of income.

Vegetable farming: favorable climatic conditions, important export product, and good source of income.

Land size: one of the most important variables, especially for grain production. The scarcity or abundance of land size is important for agricultural income.

Age, education level, place of residence, off-farm income, farm income, technical support, and neighbor support: these are important variables in terms of agricultural production.

All of the variables mentioned above directly or indirectly affect the farmer's production behavior.

### 2.3. Sociodemographic Data Form

The survey questions were rearranged using data from similar studies on this subject. The survey consists of questions related to age ($\leq$40, 41–50, 51–60, $\geq$61), education level (primary, middle school, high school, university), place of residence (rural, city), off-farm income (yes, no), agricultural branch (cereal farming, animal husbandry, fruit farming, vegetable farming, mixed farming), animal husbandry (yes, no), land size (0–5, 5–10, 10–15, 15–30, above 30), farm income (low, intermediate, high), truck use (yes, no), technical support (yes, no), neighbor support (yes, no), social support (poor, moderate, strong), and use of an operating vehicle (yes, no). All of the farmers interviewed are male and own an agricultural enterprise. Therefore, the data obtained include the characteristics of the business owner. Farm income is the level of income that a farmer earns in comparison to other farmers in his village. Equipment assets include all of the equipment used for agricultural activity on the farm. Technical support is the support received from the agricultural engineer and the veterinarian in order to continue agricultural production activity. Neighbor support consists of financing, labor, agricultural inputs, and agricultural tools and equipment.

### 2.4. Generalized Anxiety Disorder Scale (GAD-7)

Generalized anxiety disorder (GAD) is a chronic and highly prevalent disorder in the adult population. Though it occurs at a substantial frequency, the rate of GAD diagnosis is low [25]. The Generalized Anxiety Disorder (GAD-7) Scale is widely used in clinical practice and research due to its diagnostic reliability and efficiency. The GAD-7 is reported to be a valid, brief test in clinical, investigational, and general population samples [25,26]. The validity and reliability of the GAD-7 in Turkish was confirmed by Konkan et al. [25]. This scale determines the symptom frequency of the participants in the previous two weeks with a 4-point Likert scoring system (0: none, 1: a few days, 2: more than half of the days, 3: almost every day). The total score of the scale is evaluated in four categories; 0–4: minimal; 5–9: mild; 10–14: moderate; 15–21: severe. A cut-off value of 10 for the GAD-7 total score was determined to be the threshold value for a diagnosis of GAD.

### 2.5. Oslo Social Support Scale (OSSS-3)

OSSS-3 consists of three items that evaluate the level of social support. It is used for epidemiological and population-based research. Study participants were asked three questions on how many close friends they had, how involved other people were in their lives, and the availability of help from their neighbors. The response categories were assessed independently for each of the three questions, and a sum score was created by adding the three scores. The total score obtained from the scale ranges between 3 and 14 [27]. The Oslo Social Support Scale has been used in several studies, thus supporting its feasibility and predictive validity with respect to psychological distress [27,28]. This study was in accordance with the principles expressed in the Declaration of Helsinki. Gazi University Ethics Committee Number is 2021-398. Survey questions are given in Table 1.

**Table 1.** Summary of the survey design and questions.

| Section Name | Question | Question Type(s) | Possible Responses |
|---|---|---|---|
| Consent | Q1.2 | Willingness to participate | Will participate/won't |
| Demographics | Q2 | Relationship and activity with farmer organisations; age; gender; district; household size | Yes/no; amount of time; male, female, prefer not to say; age range; open-ended; household size |
| Farming Systems | Q3.1 | Which best describes your farming system | Crops, livestock, horticulture, vegetables diversified (Mixed) |
| Getting social support and technical assistance | Q4.1 | What social and technical support did you get during the COVID-19 pandemic? <br> - Have you received technical support from agricultural organizations <br> - Have you received support from neighbors or relatives for agricultural activity <br> - How easy it is to get help from your neighbors, if you need it. | What social and technical support did you get during the COVID-19 pandemic? <br> - Have you received technical support from agricultural organizations <br> - Have you received support from neighbors or relatives for agricultural activity <br> - How easy it is to get help from your neighbors, if you need it. |
| Generalized Anxiety Disorder Scale, and the Oslo Social Support Scale. | Q5.1 | Over the past two weeks, how often have you been bothered by the following problems? <br> - Are you angry, anxious, anxious <br> - Inability to control or stop your worries <br> - Don't worry too much about different things <br> - Inability to relax and relax <br> - Getting angry, angry or irritable quickly <br> - Don't be afraid that something too bad will happen | Over the past two weeks, how often have you been bothered by the following problems? <br> - Are you angry, anxious, anxious <br> - Inability to control or stop your worries <br> - Don't worry too much about different things <br> - Inability to relax and relax <br> - Getting angry, angry or irritable quickly <br> - Don't be afraid that something too bad will happen |

Source: Covidien-19 Agriculture Sector in Turkey Views on the Impact on Farmers-2020. Percentage options 0–25%, 26–50%, 51–75%, 76–100%; Used a 5-point scale (1 = Strongly Disagree → 5 = Strongly Agree). OSSS-3 and GAD-7 questions, never, a few days, more than half of the days, almost every day.

### 2.6. Data Analysis Technique

The data analysis was carried out in two stages. The first stage was determining the variables that could affect the level of anxiety using univariate analysis. Since anxiety level has an ordinal level of measurement, nonparametric tests were used in the univariate analysis. At this stage, the effects of gender, chronic discomfort, and the presence of a relative who has caught COVID-19 on the anxiety level were calculated using the Mann–Whitney U test, and the effect of the agricultural business owner education levels, place of residence, income level, and social support on anxiety level were determined using the Kruskal–Wallis H test. The second step of the analysis was to analyze the variables that were found to be important in the univariate analysis with the ordinal logistic regression model. Odds ratios (ORs) and 95% confidence intervals were used in the interpretation of

important relationships. SPSS version 23.0 software was used to analyze the data, and $p <$ 0.05 was considered statistically significant.

## 3. Results

The gender of all farmers included in the assessment is male. None of the farmers had COVID-19 in the 15 days prior to the surveys. The distribution of farmers' anxiety levels during the COVID-19 outbreak is given in Table 2. All farmers appear to have had varying degrees of anxiety.

**Table 2.** Number of farmers with different anxiety levels (n = 2125).

| Anxiety Level | Frequency | Percent |
|---|---|---|
| Normal | - | - |
| Minimal | 958 | 45.1 |
| Moderate | 1111 | 52.3 |
| Severe | 56 | 2.6 |
| Total | 2125 | 100.0 |

The univariate analysis results between the anxiety level and other variables in the study and descriptive statistics such as frequency, percentage, mean, and standard deviation are shown in Table 3. As seen in the table, age, place of residence, income status, field of activity, use of tractors and trucks, availability of livestock, technical support from agricultural establishments during the COVID-19 outbreak, support from neighbors during the COVID-19 outbreak, and social support had significant effects on anxiety ($p < 0.05$). The results show that as the age decreased, the rate of severe anxiety in farmers increased. Furthermore, 70.3% of the farmers lived in rural areas, and the rate of severe anxiety in these farmers was about twice as high as that of urban farmers. Further, the rate of severe anxiety increased as the income level increased.

When the fields of activity were analyzed, the farmers engaged in animal production or vegetable growing were observed to have higher anxiety rates than other farmers. The decrease in rate of severe anxiety was remarkable as the land size increased. Regarding operating vehicles, farmers who had a tractor or truck had a higher rate of severe anxiety. The incidence of rate of severe anxiety was found to be lower in animal husbandry than in non-farmers, but the moderate anxiety rate was quite high in animal husbandry (72%). The results of similar studies on this subject are in line with our findings [29–33].

In addition, the level of anxiety was lower in farmers who received support from agricultural institutions or their neighbors during the COVID-19 outbreak. Small businesses—agricultural enterprises that have less than 10.0 hectares of land—make up 80.7% of agricultural enterprises in Turkey. The fact that small enterprises are prevalent may be the main reason for the lower level of anxiety in these farmers because the farmers in this group have a high solidarity effort of social assistance.

Similarly, as social support increased, the anxiety level decreased. However, as reported in other studies, travel restrictions may lead to labor shortages in critical sectors like agriculture that are dominated by migrant workers [34–36]. Timilsina et al. [37] stated that the government should provide support (quality seed, fertilizer, direct financial support, etc.) to vulnerable farmers to increase the resilience of the agricultural sector during the pandemic.

**Table 3.** Univariate analysis of farmers' anxiety about the pandemic.

| Variables | Mild | Moderate | Severe | Total | Statistics | P |
|---|---|---|---|---|---|---|
| | | | **Anxiety Level** | | | |
| Socioeconomic Conditions of Farmers Age | | | | | 19.100 [b] | <0.001 |
| ≤40 | 208 (44.7) | 234 (50.3) | 23 (5.0) | 465 (21.9) | | |
| 41–50 | 256 (38.8) | 386 (58.6) | 17 (2.6) | 659 (31.0) | | |
| 51–60 | 315 (50.5) | 296 (47.4) | 13 (2.1) | 624 (29.4) | | |
| ≥61 | 179 (47.5) | 195 (51.7) | 3 (0.8) | 377 (17.7) | | |
| Education level | | | | | 6.777 [b] | 0.079 |
| Primary | 307 (48.0) | 315 (49.3) | 17 (2.7) | 639 (30.1) | | |
| Middle school | 264 (46.1) | 301 (52.5) | 8 (1.4) | 573 (27.0) | | |
| High school | 282 (41.9) | 363 (53.9) | 28 (4.2) | 673 (31.7) | | |
| University | 105 (43.8) | 132 (55.0) | 3 (1.3) | 240 (11.3) | | |
| Place of residence | | | | | −4.508 [a] | <0.001 |
| Rural | 628 (42.0) | 820 (54.9) | 46 (3.1) | 1494 (70.3) | | |
| City | 330 (52.3) | 291 (46.1) | 10 (1.6) | 631 (29.7) | | |
| Off-farm income | | | | | −0.371 [a] | 0.710 |
| Yes | 562 (45.8) | 623 (50.8) | 41 (3.3) | 1226 (57.7) | | |
| No | 396 (44.0) | 488 (54.3) | 15 (1.7) | 899 (42.3) | | |

**Table 3.** *Cont.*

| | | Anxiety Level | | | | |
|---|---|---|---|---|---|---|
| **Variables** | **Mild** | **Moderate** | **Severe** | **Total** | **Statistics** | ***P*** |
| Farm income | | | | | 85.937 [b] | <0.001 |
| Low | 183 (61.2) | 116 (38.8) | 0 (0.0) | 299 (14.1) | | |
| Intermediate | 648 (47.0) | 692 (50.2) | 39 (2.8) | 1379 (64.9) | | |
| High | 127 (28.4) | 303 (67.8) | 17 (3.8) | 447 (21.0) | | |
| Farm Characteristic-sAgricultural Branch | | | | | 636.053 [b] | <0.001 |
| Cereal farming | 539 (83.3) | 105 (16.2) | 3 (0.5) | 647 (30.4) | | |
| Animal Husbandry | 6 (4.3) | 125 (89.3) | 9 (6.4) | 140 (6.6) | | |
| Fruit farming | 214 (38.9) | 328 (59.6) | 8 (1.5) | 550 (25.9) | | |
| Vegetable farming | 48 (15.8) | 222 (73.0) | 34 (11.2) | 304 (14.3) | | |
| Mixed farming | 151 (31.2) | 331 (68.4) | 2 (0.4) | 484 (22.8) | | |
| Land size (Hectares) | | | | | 20.498 [b] | <0.001 |
| (0, 5] | 338 (43.4) | 403 (51.8) | 37 (4.8) | 778 (36.6) | | |
| (5, 10] | 265 (41.9) | 354 (56.0) | 13 (2.1) | 632 (29.7) | | |
| (10, 15] | 133 (45.4) | 157 (53.6) | 3 (1.0) | 293 (13.8) | | |
| (15, 30] | 120 (48.4) | 125 (50.4) | 3 (1.2) | 248 (11.7) | | |
| (30, →] | 102 (58.6) | 72 (41.4) | 0 (0.0) | 174 (8.2) | | |

**Table 3.** *Cont.*

| Variables | Anxiety Level | | | | | |
|---|---|---|---|---|---|---|
| | Mild | Moderate | Severe | Total | Statistics | P |
| Use of a tractor | | | | | −3.240 [a] | 0.001 |
| Yes | 825 (44.0) | 996 (53.1) | 56 (3.0) | 1877 (88.4) | | |
| No | 133 (53.6) | 115 (46.4) | 0 (0.0) | 248 (11.6) | | |
| Truck use | | | | | −17.997 [a] | <0.001 |
| Yes | 234 (24.2) | 682 (70.7) | 49 (5.1) | 965 (45.4) | | |
| No | 724 (62.4) | 429 (37.0) | 7 (0.6) | 1160 (54.6) | | |
| Equipment use | | | | | −1.396 [a] | 0.163 |
| Yes | 685 (44.0) | 833 (53.5) | 38 (2.5) | 1556 (73.2) | | |
| No | 273 (48.0) | 278 (48.9) | 18 (3.1) | 569 (26.8) | | |
| Animal husbandry | | | | | −11.440 [a] | <0.001 |
| Yes | 180 (26.1) | 497 (72.0) | 13 (1.9) | 690 (32.5) | | |
| No | 778 (54.2) | 614 (42.8) | 43 (3.0) | 1435 (67.5) | | |
| Technical support | | | | | −6.758 [a] | <0.001 |
| Yes | 639 (50.3) | 622 (48.9) | 10 (0.8) | 1271 (59.8) | | |
| No | 319 (37.4) | 489 (57.3) | 46 (5.4) | 854 (40.2) | | |
| Neighbor support | | | | | −6.459 [a] | <0.001 |
| Yes | 241 (38.8) | 366 (58.9) | 14 (2.3) | 621 (29.2) | | |
| No | 507 (33.7) | 925 (61.5) | 72 (4.8) | 1504 (70.8) | | |

**Table 3.** *Cont.*

| Variables | Anxiety Level | | | | Statistics | *P* |
|---|---|---|---|---|---|---|
| | **Mild** | **Moderate** | **Severe** | **Total** | | |
| Social Support | | | | | 18.854 [b] | <0.001 |
| Poor | 216 (39.1) | 303 (54.8) | 34 (6.1) | 553 (26.1) | | |
| Moderate | 519 (47.1) | 561 (50.9) | 22 (2.0) | 1102 (52.0) | | |
| Strong | 223 (48.1) | 241 (51.9) | 0 (0.0) | 464 (21.9) | | |

[a] Mann—Whitney test. [b] Kruskal—Wallis test.

The relationships between variables found to be important in the univariate analysis and the level of anxiety were examined using multivariate analysis. The chi-square value obtained for the parallel line's assumption of ordinal logistic regression model was 8.925 ($p$ = 0.989, df = 25, chi-square critical value = 37.653). According to this result, the ordinal logistic regression model showed a good fit with the observed values. Odds Ratio (OR) values and significance levels obtained from the ordinal logistic regression model are given in Table 4. The level of anxiety of farmers between the ages of 41 and 50 years was higher than that of the farmers aged 61 years and over (OR = 1.452, 95% CI: 1.070–1.969). In contrast to rural areas, living in urban areas was a protective factor against the anxiety that the farmers experienced (OR = 1.556, 95% CI: 1.207–2.006). Moreover, anxiety was higher in high-income farmers than in other farmers. The anxiety of farmers engaged in animal production (OR = 3.017, 95% CI: 1.766–5.153) and vegetable farming (OR = 4.222, 95% CI: 2.695–6.613) was higher than that of farmers in mixed production. The anxiety of farmers who produced plants was lower than that of farmers who were engaged in mixed production (OR = 0.162, 95% CI: 0.110–0.238). Hai-Ying and Chang-Wei [38] determined that the market risks of vegetable production have increased significantly. The COVID-19 pandemic has impacted almost all stages of the vegetable supply chain but has had a greater impact on the sales stage. Kumar et al. [39] reported that during the implementation of COVID-19 restrictions, horticultural farmers faced greater problems than other farmers. Vegetable and fruit producers have stated that they are in a great panic due to a lack of labor, a lack of demand, and transport bottlenecks that completely disturb the market linkage [40]. Richards and Rickard [41] reported that Canadian fruit and vegetable markets were significantly impacted by the spread of COVID-19. From the agricultural production point of view, Marwanti and Antriandarti [15] reported that in order to reduce farmers' anxiety, governments should support them at all stages, from the supply of agricultural inputs to the marketing stage.

The anxiety levels of farmers with a land of 30 hectares or less were higher than those of farmers with a land of more than 300 hectares. Anxiety was higher in truck farmers (OR = 2.117, 95% CI: 1.615–2.776) and animal husbandry (OR = 1.700, 95% CI: 1.232–2.345). It was determined that 87.56% of the farmers engaged in animal husbandry owned trucks. Therefore, it would be reasonable to consider truck owners and animal husbandry to be in the same category. Obtaining technical support from agricultural organizations (OR = 0.452, 95% CI: 0.357–0.574), receiving neighbor support (OR = 0.707, 95% CI: 0.558–0.896), and having strong social support were protective factors against anxiety.

**Table 4.** Ordinal logistic regression analysis of factors influencing farmers' anxiety.

| Variables | OR | SE | *p* | OR (95%CI) |
|---|---|---|---|---|
| Socioeconomic Conditions of Farmers Age | | | | |
| ≤40 | 1.017 | 0.172 | 0.922 | (0.726, 1.426) |
| 41–50 | 1.452 | 0.156 | 0.017 | (1.070, 1.969) |
| 51–60 | 0.917 | 0.156 | 0.580 | (0.676, 1.245) |
| ≥61 [a] | - | - | - | - |
| Place of residence | | | | |
| Rural | 1.556 | 0.130 | 0.001 | (1.207, 2.006) |
| City [a] | - | - | - | - |
| Farm income | | | | |
| Low | 0.167 | 0.234 | <0.001 | (0.106, 0.265) |
| Intermediate | 0.333 | 0.170 | <0.001 | (0.239, 0.465) |
| High [a] | - | - | - | - |
| Farm Characteristics Agricultural Branch | | | | |
| Cereal farming | 0.162 | 0.197 | <0.001 | (0.110, 0.238) |
| Animal Husbandry | 3.017 | 0.273 | <0.001 | (1.766, 5.153) |
| Fruit farming | 1.350 | 0.192 | 0.118 | (0.927, 1.968) |
| Vegetable farming | 4.222 | 0.229 | <0.001 | (2.695, 6.613) |
| Mixed farming | - | - | - | - |
| Land size (hectares) | | | | |
| (0, 5] | 1.808 | 0.273 | 0.030 | (1.060, 3.084) |
| (5, 10] | 2.787 | 0.249 | <0.001 | (1.710, 4.541) |
| (10, 15] | 1.917 | 0.256 | 0.011 | (1.160, 3.166) |
| (15, 30] | 1.848 | 0.260 | 0.018 | (1.109, 3.077) |
| (30, →] [a] | - | - | - | - |
| Use of a tractor | | | | |
| Yes | 0.817 | 0.188 | 0.284 | (0.565, 1.182) |
| No [a] | - | - | - | - |
| Truck use | | | | |
| Yes | 2.117 | 0.138 | <0.001 | (1.615, 2.776) |
| No [a] | - | - | - | - |
| Animal husbandry | | | | |
| Yes | 1.700 | 0.164 | 0.001 | (1.232, 2.345) |
| No [a] | - | - | - | - |
| Technical support | | | | |
| Yes | 0.452 | 0.121 | <0.001 | (0.357, 0.574) |
| No [a] | - | - | - | - |
| Neighbor support | | | | |
| Yes | 0.707 | 0.121 | 0.004 | (0.558, 0.896) |
| No [a] | - | - | - | - |
| Social Support | | | | |
| Poor | 1.522 | 0.171 | 0.014 | (1.088, 2.130) |
| Moderate | 1.111 | 0.146 | 0.471 | (0.835, 1.478) |
| Strong [a] | - | - | - | - |

SE, Std. Error; OR, Odds ratio; CI, Confidence interval. [a] reference groups.

## 4. Discussion

Anxiety disorders and other psychiatric diseases are known to be more common in farmers than in the general population, even when there is no danger of an epidemic disease [11,42]. In addition to these factors, farmers who are already in poor economic situations were seriously affected as a result of disruptions due to COVID-19 measures during the harvest season, difficulties in labor supply, and disconnection of the supply chain and the market [10,43]. Unfortunately, farmers who primarily live-in rural areas have limited access to mental health services [10]. The continuation of agricultural production and the mental health of farmers working in the field are important for people's nutrition and their survival. Our study made it possible to examine the anxiety disorders experienced by farmers during the pandemic and the factors affecting them in terms of socioeconomic and social support.

More than half of the farmers were found to have at least moderate anxiety. according to the GAD 7 results (total GAD-7 score ≥10. Rudolphi et al. [11] examined 170 farmers with the GAD-7 scale and found that 71% of the farmers had different levels of anxiety. Considering that the COVID-19 outbreak had not started at the time of their study, it can be said that our results are in line with the literature. According to the results, as age increases, anxiety levels tend to decrease. Ahearn noted in his study that young farmers experience more stress than their more experienced colleagues [44]. Chronic stress can cause mental health disorders and especially contributes to the development and progression of depression and anxiety [45].

Rudolphi et al. [11] stated that the prevalence of depression and anxiety disorder in young farmers between the ages of 18 and 37 is higher than that in the general population. The mental health of young farmers should also be taken into consideration to ensure a sustainable workforce. A statistically significant correlation was found between anxiety levels and variables such as the farmers' place of residence, income status, field of activity, land availability, the use of a truck, and the use of tractors.

Farmers living in rural areas have higher anxiety levels than those living in urban areas. As Patnaik stated in his study, this may be due to the imbalance of economic, cultural, and educational resources between urban and rural areas [43]. This could also be attributed to the fact that farmers who resided in rural regions during the epidemic had fewer alternatives to make a living. In our study, the anxiety level was found to be higher in farmers with high income compared to other farmers. This result may be due to the fact that high-income farmers have invested more capital in agricultural production than other farmers. As the return on investment decreases and is delayed, the ability to continue production activity decreases. Farmers in this group who invest more in changes in the marketing network or in prices are affected [46]. In addition, producers in the high-income group employ more workers. During the pandemic period, the management of workers may have become difficult, and the anxiety level of farmers may have increased. Since the production area of low- and middle-income farmers is small, they can harvest their products with the help of their own labor force and neighbors. This may have caused low- and middle-income farmers to experience less anxiety.

In our study, farmers are divided into five groups according to their field of activity: animal husbandry, vegetable farming, fruit farming, mixed farming, and cereal production. The level of anxiety of farmers engaged in animal production or vegetable production alone was found to be higher than those who engaged in mixed production.

In agricultural enterprises, as the product variety increases, the risk factor decreases. For this reason, the anxiety level may be found to be higher only in animal husbandry and vegetable farming. An enterprise engaged in animal husbandry or vegetable farming alone is faced with complete extinction or an inability to meet production costs during an adverse event in this area. This means that the producer leaves this production area completely or reduces production. Many producers prefer mixed production according to agricultural production conditions in order to reduce risks. On the other hand, the closure of hotels, restaurants, and other food-providing places rendered farmers unable to sell

certain products and forced them to sell these goods at low prices. This situation mostly affected producers engaged in vegetable farming and animal husbandry.

The anxiety level of farmers in animal husbandry was found to be higher than that of the other groups. This may be due to the requirement to bring animal products to the market in a shorter time and the fact that many marketplaces are closed during the pandemic period. In addition, animal husbandry has had difficulties in selling and slaughtering animals. The majority of farmers engaged in animal husbandry in Turkey raise their animals to market for the Feast of the Kurban. In the period during which the surveys were conducted, the short time of two months to the Kurban Feast and the delay in announcing the measures to be taken for animal transport and sale may have increased the level of concern of farmers. The anxiety levels of farmers engaged in cereal production were found to be lower than those of mixed farmers. Since the harvest and marketing of crop production enterprises (wheat, barley, corn, rye, one-year fodder, etc.) do not coincide with the period when the pandemic was intense, their anxiety levels may have remained at a lower level. Statements made by authorities that the pandemic period will end or decrease in June and that everything will return to normal may have alleviated the concerns of these producers.

Farmers who have over 30 hectares of land have lower anxiety levels than other farmers. This may be due to the fact that farmers with little land are concentrated in a single field of activity such as vegetable and fruit farming. As the land asset increases, the possibility of mixed production increases and, as discussed above, the risk factor decreases as the product variety increases. The anxiety level was observed to be higher in farmers who own trucks. This may be a result of the disruption of the supply chain due to the implementation of curfews and the prohibition of intercity travel to prevent the spread of the epidemic. Farmers who are facing economic difficulties may try to contribute to their budgets with product transportation.

The social support system is an important resource for solving and preventing people's psychological problems and protecting their mental health [47,48]. The OSSS-3 scale was used in our study to examine the effect of social support on anxiety. According to the OSSS-3 total scores, farmers with strong social support had lower anxiety levels than farmers with less social support. The specific conditions of agriculture in every country require that this sector be protected and supported directly by the state or through institutions authorized by the state [49]. Farmers were asked whether they received technical support from agricultural organizations during the COVID-19 outbreak and whether they would receive support from their neighbors. The results show that getting support from agricultural organizations or neighbors is associated with low anxiety levels. These results show that farmers should be supported by agricultural organizations and the state, especially in technical matters. According to the research that we found, no survey study examined the psychological effects of the COVID-19 epidemic on farmers while our study was being prepared. Other strengths of our study are the size of the sample and the assessment of anxiety levels by a psychiatrist. The disadvantage of our research is that it is a cross-sectional, longitudinal follow-up study. Face-to-face interviews under examination room conditions and the evaluation of the gender factor will be beneficial for the generalizability of future studies and for overcoming limitations.

To prevent the rapid rise in agricultural input prices, facilitating border procedures in essential inputs such as fertilizers, veterinary medicines, and pesticides by allowing for digital copies of certificates could be beneficial. Quanyson et al. [50] pointed out that digital transformation may be promising for improving conditions for vulnerable farmers even in a period of crisis such as the COVID-19 pandemic.

In order to reduce the anxiety of farmers in animal husbandry, the government should determine the locations of mass slaughterhouses to optimize the supply chain and meet consumer demand. In addition, it should ensure that the distribution of slaughtered animals is maintained and that the suppliers and distributors in this regard are managed through an appropriate channel.

The realization of this situation may contribute to reducing the anxiety levels of animal producers and to their production. Facilitating and promoting widespread access to animal feed and other raw materials at the country level and contributing to their distribution will reduce the level of concern for animal husbandry. This will contribute to the sustainability of animal production. According to a similar study on this subject, dairy farmers require more directed and targeted support, as their pandemic-related feedback stock and market-access hitches are serious [51].

In order to reduce the anxiety levels of fruit and vegetable farmers, it is necessary to store products in production areas. The delivery of products taken to warehouses and to consumers through supply chains will enable communication between the producer and the consumer in a shorter time. With the realization of this situation, the anxiety levels of fruit and vegetable farmers will be reduced, preventing a decrease in or the cessation of production.

In this study, farmers in vegetable, fruit, and animal production branches were found to be more psychologically vulnerable and had higher levels of anxiety during the COVID-19 outbreak than farmers in other agricultural branches. In particular, specific agricultural support policies can be implemented for those farmers who are heavily affected by the COVID-19 outbreak. Therefore, more studies are needed for farmers in vegetable, fruit, and animal production branches.

## 5. Conclusions

Employees working in agricultural production are of great importance for the nutrition and economy of the country's population. In our study, the anxiety levels of farmers during the COVID-19 outbreak were examined and the affecting socioeconomic and social support factors were determined. The COVID-19 pandemic has created different levels of anxiety in farmers in terms of different branches of agricultural production. Taking these results into consideration is important for preventing problems in agricultural production and therefore in access to food. As it is not yet possible to eradicate the epidemic, institutional measures should be taken to alleviate socioeconomic challenges. Access to mental health services may be facilitated for farmers at risk for serious psychiatric illnesses or even suicide. Except for the livestock industry with the production of fruits and vegetables, the measures taken for the COVID-19 outbreak in Turkey have not caused significant problems for agricultural production in other branches across the country. This situation has likely arisen as a result of the agricultural support policies announced by the government. New incentives provided by the government to agricultural branches (vegetables, fruits, animal husbandry) with a high level of anxiety are important for agricultural production. If new incentives are not provided, farmers may turn to production branches that are associated with lower levels of concern. At present, it is of great importance to evaluate the possible consequences of the COVID-19 epidemic on all agriculture-related production branches. As far as we know, there are no studies on this subject in the literature to date. Our study will not only help policymakers create effective policies but also help to minimize the negative impact on agricultural production branches in the event of similar outbreaks in the future. This study on the agricultural industry in Turkey is expected to serve as a warning or advice to other countries, including developing countries, so that they can maintain production, particularly in the agricultural industry, during severe epidemics.

**Author Contributions:** Conceptualization, C.C. and B.A.; methodology, C.C., B.A. and M.G.; software, B.A.; validation, C.C. and B.A.; formal analysis, B.A.; writing—original draft preparation, C.C., B.A. and M.G.; writing—review and editing, C.C., B.A. and M.G. All authors have read and agreed to the published version of the manuscript.

**Funding:** This research received no external funding.

**Institutional Review Board Statement:** The study was conducted according to the guidelines of the Declaration of Helsinki, and approved by Gazi University Ethics Committee (protocol code 2021-398 and date of approval E-77082166-604.01.02-70583).

**Informed Consent Statement:** Informed consent was obtained from all subjects involved in the study

**Data Availability Statement:** Data sharing not applicable.

**Acknowledgments:** The authors thank four anonymous reviewers for their helpful comments and suggestions. We thank all farmers in the agricultural production branch, agricultural institutions, and organizations in Turkey from which we received the primary data of the study.

**Conflicts of Interest:** The authors declare no conflict of interest.

## Abbreviations

COVID-19　Coronavirus Disease 2019
GAD-7　　　Generalized Anxiety Disorder Scale
OSSS-3　　 Oslo Social Support Scale
WHO　　　 World Health Organization

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
