# Peer review of "Impacts of COVID-19 on Agricultural Production Branches: An Investigation of Anxiety Disorders among Farmers"

_sustainability, doi:10.3390/su13095186_

Round 1
Reviewer 1 Report
In the manuscript the authors presented an important point, although the manuscript has some drawbacks.
Main remarks:
1. Intruduction - in the introduction, please write about sustainability. The manuscript was sent to Sustainability.
2. Literature review - where is? This part is missing from the manuscript. Please write about agricultural development, rural areas, rural development. Suggested publications:
- Roman, M.; Roman, M.; Prus, P.; Szczepanek, M. Tourism Competitiveness of Rural Areas: Evidence from a Region in Poland. Agriculture 2020, 10, 569. https://doi.org/10.3390/agriculture10110569
- Farrell, L.J.; Kenyon, P.R.; Morris, S.T.; Tozer, P.R. The Impact of Hogget and Mature Flock Reproductive Success on Sheep Farm Productivity. Agriculture 2020, 10, 566. https://doi.org/10.3390/agriculture10110566
3. Methods, results ... e.t.c. - in my opinion, there are no taxonomic and statistical methods, e.g. proposition - own measure of agricultural development or selected production.
4. Conclusions - very short. In conclusions, please also answer the following questions:
• what are the research gaps?
• what is new to this manuscript?
Author Response
I would like to thank the esteemed referees for their valuable insight and constructive contribution.

Reviewer 2 Report
Thank you for the opportunity to review. This is an important topic and I'm excited about the final manuscript when ready. Here are some suggestions to improve upon the paper as it stands today:
- Please be consistent with "COVID-19" (no lowercase).
- The manuscript needs significant English/grammar review (verb conjugations, etc).
- Please correct comma splices, and/or using commas were periods are needed.
- the last 2 paragraphs under section 2.2 should probably be moved to results.
- there needs to be more discussion on the scales (section 2.4 and 2.5) regarding why they were chosen and fit within the study.
- please use the lower-case "p" for p-value (not "P")
- recommend breaking-up the data in Table 2 based on variable-type or other common factor to make it a little easier to interpret (example: demos, farm characteristics, etc).
- ...same for Table 3.
- I'm not locating the critical value for the X2 result?
- a lot of grammar issues - first line of page 8 (example).
- please consider inserting sub-headings/sub-sections under the discussion section to help with readability.
- the last sentence of page 7 seems to contradict the title/premise of the article. I may be interpreting it incorrectly.
- please list/include the IRB information for this study (human subjects).
- back to section 2.2 - please discuss each variable in further detail by addressing how/why it is included in the study.
The manuscript is promising. I strongly recommend the use of additional sub-headings to keep the message on-track and assist with readability. Please conclude with a stronger linkage between the study and potential ramifications in the farming industry in Turkey (beyond farmers deciding to seek work in other farming industry segments).
Author Response

(The authors gave the same response as above.)

Reviewer 3 Report
The topic of the paper is interesting, and the paper is well structured. However, the paper needs careful re-reading, and some revisions.
Affiliations
- The third affiliation is not associated with any author
Abstract
- “… examine their impact …” should be “… examine its impact …” (if you are referring to the level of anxiety)
- “… branches, Generalized …” should be “… branches. Generalized …”
- “Age, place of residence, … level, it was found …” should be “Age, place of residence, … level were found …”
Introduction
- Please improve the logical flow of the information provided
- Rewrite the sentence “After the first report …”. E.g.: After the first report on March 11, 2020, various health and safety precautions have been taken in Turkey in order to prevent the spread of the disease
- In the sentence “It is known that agricultural production is a sector with a high-risk rate …”, what risk are you referring to?
- In the sentence “In a study examining a single case [9], they investigated …”, replace “they” with “the authors”
- Rewrite the sentence “In this study before the COVID-19 outbreak [10] found …”. E.g.: Moreover, prior to COVID-19 outbreak, a study [10] found ...”
- In the sentence “In this context, the aim of this study …”, remove “of anxiety disorders”
Section 2.1
- Were the preventive measures the same in all provinces?
- In the sentence “Data were collected …, and it was …”, explain the second subject (it) because it is different from the first one
Section 2.2
- Correct the sentence “The questionnaire contained structured consisted of …”
- Replace “… (2) Production changes …” with “… (2) production changes …”
- Rewrite the sentence “Survey interview, after the first start …”
- Regarding the sentence “Interviews were conducted in the agricultural farmers surveyed, perceived to have the knowledge, experience and perspective required to pro-vide information on impacts in agricultural businesses that make management decisions”, was perceived knowledge, experience, and perspective a criterion for selecting farmers to interview? If yes, how did you evaluate that perception? And why did you write in Section 2.1 that farmers were randomly selected? If not, what is the meaning of that sentence?
- Remove “Characteristics of agricultural enterprises in Turkey;” and “Production branches of farmers in the work areas;”
Section 2.3
- Make the subject of the sentence “It was prepared …” explicit
- What literature are you referring to in the sentence “It was prepared …”?
- Also mention the education level and other variables investigated among the questions. In addition, specify the possible answers for each question (e.g. age ≤ 40, 41-50, 51-60, ≥ 61)
- About the education level, is this level referred to father and mother? What is the expected response if parents have a different educational level? Why are you interested in the level of education of the parents and not that of the interviewee?
- About farm income, what do low, intermediate, and high mean?
- About equipment presence, what equipment do you consider?
- About technical support, what does it consist of?
- About neighbor support, what does it consist of? (e.g.: financial? psychological?). Why did you consider this variable since it seems to be included in the OSSS-3?
Section 2.4
- Make the subject of the sentence “This scale has …” explicit
- Replace “This scale determined …” with “This scale determines …”
- Move “(0: none, …)” after “… Likert scaling”
- Spitzer et al. [17] defined the score 0-4 as minimal
Section 2.5
- Remove “with high values representing strong, low values representing weak levels of social support” in order to avoid unnecessary repetition with the next sentence
Section 2.6
- Here chronic discomfort and the presence of a relative who has caught COVID-19 are mentioned, but they no appear in Table 2
- Remove “The effect of place of residence, income level and social support on anxiety level was analyzed with the Kruskal-Wallis H test.” in order to avoid unnecessary repetition with the previous sentence
Section 3
- Remove “The proportion of farmers with severe anxiety seems to be 2.6%. The proportion of farmers with moderate anxiety is as high as 52.3%.” in order to avoid unnecessary repetition with Table 1
- Also add the normal (or minimal) level in Table 1, even if the frequency and percentage are equal to 0
- The phrases "level of severe anxiety" and "severe anxiety level" are ambiguous. I suggest referring to the rate
- Remove the comma between subject and verb in the sentence “80.7 % of agricultural enterprises in Turkey, is the group …”
- You state that “the farmers in this group have a high sense of social assistance”. Have you analysed the relationship between variables? If so, it would be interesting to report the data
- You use “however” between the following sentences: “Similarly, anxiety level decreased as social support increased” and “in other studies, Travel restrictions may lead to labor shortages in critical sectors like agriculture that are dominated by migrant workers”. Why? What is the connection between the two sentences?
- Replace “… Travel restrictions …” with “… travel restrictions …”
- The sentence “Timilsina et al. [30] suggested that …” is not clear
- Did you aggregate the responses by province? If so, do any differences emerge?
- The comments on the results of the ordinal logistic regression analysis appear similar to those of the univariate analysis. Are there any additional significant findings?
- The text “Hai-ying and Chang-wei … inputs to the marketing stage” could support the discussion, not the results
- Replace “sales stag” with “sales stage”
- Format the superscripts in Table 3
Discussion
- Please explain the sentence “It was found that all the farmers participating in the study met the criteria of the GAD-7 scale (GAD-7 total score 5)”
- Please explain T.9 in the sentence “More than half of the farmers (T.9) have moderate anxiety”
- What literature are you referring to in the sentence “… it can be said that our results are in line with the literature”?
- Rewrite the sentence “According to the results of the multivariate analysis; It has been determined that as age increases, anxiety levels tend to decrease”
- Replace “… production increases and as discussed above, the risk factor …” with “… production increases and, as discussed above, the risk factor …”
- Why do you use "however" between the sentences “The realization of this situation may contribute to reducing the anxiety levels of ani-mal producers and to their production” and “facilitating and promoting widespread access to animal feed and other raw materials …”?
- Replace “Dairy farmers” with “dairy farmers”
- Remove “will be prevented” in the sentence “… preventing the decrease in production or cessation of production will be prevented”
- Please provide more practical and clear suggestions for future studies
Conclusions
- In the sentence “The measures taken for Covidien-19 restrictions …”, what does “Covidien-19” mean?
- The sentences “Access to mental health services should be…” and “If new incentives are not provided, it is predicted …” are not derived from results and discussion
- Please make conclusions more consistent with previous sections
References
- The number 2 has two links. Why?
Author Response

(The authors gave the same response as above.)

Reviewer 4 Report
The study explored the level of anxiety during the Covid-19 outbreak in different agricultural branches of farmers in Turkey during the COVID-19 outbreak and to examined their socio-economic impact on this population. I would like to stress out that I support the potential publication of this paper due to its scientific interest. However, I see a few issues that ought to be addressed from my perspective.
The authors need to expand the review of literature that is relevant to their study. I encaourage to include and discuss the following articles in the "Introduction" section :
Darnhofer, I. (2020). Farm resilience in the face of the unexpected: Lessons from the COVID-19 pandemic. Agriculture and Human Values, 37, 605-606.
Darnhofer, I. (2021). Farming Resilience: From Maintaining States towards Shaping Transformative Change Processes. Sustainability, 13(6), 3387.
Bochtis, D., Benos, L., Lampridi, M., Marinoudi, V., Pearson, S., & Sørensen, C. G. (2020). Agricultural workforce crisis in light of the COVID-19 pandemic. Sustainability, 12(19), 8212.
Moreover, the goal of the study needs to be properly highlighted and justified. Instead of setting their aim in the frame of a simple question, I would recommend that the authors attempt to present the key objectives of their study with regards to what is presently known (i.e. literature), thus highlighting the added value of the article.
I recommend the authors add hypotheses and/or state them in a more explorative way as research questions.
Could the authors please add information on how the participants were recruited?
"The questionnaire was developed in consultation with epidemiologists, behavioural and public health scientists..." Could the authors please add information on how the questionnaire was developped?
Please add the number of items in the questionnaire, examples of items and the response options of the scales.
I think it is important to inform that "this study was in accordance with the principles expressed in the Declaration of Helsinki."
I recommend the author review the English of this manuscript and reformulate tables to be more clear to the audience
Author Response
Thank you very much. You can find all corrections as an attachment file.

Round 2
Reviewer 1 Report
Accept in present form. Good luck!
Author Response

(The authors gave the same response as above.)

Reviewer 2 Report
I see the manuscript has been edited for grammar significantly. Thank you.
The authors defined variables in section 2, yet did not do a suitable job discussing how/why they were included in the study (original request).
I am surprised to see many requests for changes/edits for clarity not changed by the authors. The lack of IRB approval for a study focused on human subjects is inappropriate (regardless if/how consent was obtained by study participants). This needed to be completed before the study was conducted and it is apparent based on the response comments it was omitted.
Author Response

(The authors gave the same response as above.)

Reviewer 3 Report
I have appreciated the reviews made to satisfy my comments, but some observations were not fully or properly addressed.
- Education level: Why are you interested in the level of education of the parents and not that of the interviewee? What is the expected response if parents have a different educational level?
- Data and survey: You listed “cereal farming”, “Cereal animal farming”, “Fruit farming”, and “Vegetable farming” as variables examined in the study, but they are not variables (the related variable is the Agricultural branch).
- GAD-7: Since you have corrected the GAD categories (i.e., 0-4: minimal), the sentence " Spitzer et al [21] defined a score of 0-4 as minimal." is unnecessary. Furthermore, the first row of Table 1 should refer to "Minimal".
- Data analysis: Why are chronic discomfort and the presence of a relative who has caught COVID-19 mentioned in Section 2.6 and not shown in Table 2?
- Results: I suggest replacing "level of severe anxiety" and "severe anxiety level" with “rate of severe anxiety”.
- Results: You state that “The fact that small enterprises are prevalent may be the main reason for the lower level of anxiety in these farmers because the farmers in this group have a high sense of social assistance”. On what information is the statement " the farmers in this group have a high sense of social assistance" based?
- Discussion: The sentence “More than half of the farmers were found to have moderate anxiety. according to the GAD 7 results (GAD-7 total score of ≥5).” should be replaced with “More than half of the farmers were found to have at least moderate anxiety. according to the GAD 7 results (total GAD-7 score ≥10).”
Author Response

(The authors gave the same response as above.)

Reviewer 4 Report
The paper is now suitable for publication
Round 3
Reviewer 2 Report
Please re-address request #1. "Covid-19" still present in the manuscript edits (ex. page 2).
The last section of 2.2 is very limited in information and simply listed as line items in the manuscript, versus a table or other informative presentation of variables. The last sentence is an important part of the initial request to further support/justify use of such variables - but is not cited or supported in any way. To make this manuscript better, I would recommend a little more attention in this section (presentation and more information on variables).
Thank you for including the IRB information in the manuscript. Again, method of consent is irrelevant if an ethics cmte approval was not correctly obtained beforehand for the survey.